# Multifunctional 3D-Printable Photocurable Elastomer with Self-Healing Capability Derived from Waste Cooking Oil

**DOI:** 10.3390/molecules30081824

**Published:** 2025-04-18

**Authors:** Pengyu Wang, Jiahui Sun, Mengyu Liu, Chuanyang Tang, Yang Yang, Guanzhi Ding, Qing Liu, Shuoping Chen

**Affiliations:** 1College of Materials Science and Engineering, Guilin University of Technology, Guilin 541004, China; 2120220368@glut.edu.cn (P.W.); 2120210291@glut.edu.cn (M.L.); 2120230405@glut.edu.cn (C.T.); 1020230205@glut.edu.cn (Y.Y.); 1020220195@glut.edu.cn (G.D.); 2120220336@glut.edu.cn (Q.L.); 2School of Chemistry, South China Normal University, Guangzhou 510006, China; 13355357764@163.com

**Keywords:** 3D-printable, photocurable elastomer, waste cooking oil, self-healing, multifunctional

## Abstract

This study presents a sustainable approach to transform waste cooking oil (WCO) into a multifunctional 3D-printable photocurable elastomer with integrated self-healing capabilities. A linear monomer, WCO-based methacrylate fatty acid ethyl ester (WMFAEE), was synthesized via a sequential strategy of transesterification, epoxidation, and ring-opening esterification. By copolymerizing WMFAEE with hydroxypropyl acrylate (HPA), a novel photocurable elastomer was developed, which could be amenable to molding using an LCD light-curing 3D printer. The resulting WMFAEE-HPA elastomer exhibits exceptional mechanical flexibility (elongation at break: 645.09%) and autonomous room-temperature self-healing properties, achieving 57.82% recovery of elongation after 24 h at 25 °C. Furthermore, the material demonstrates weldability (19.97% retained elongation after 12 h at 80 °C) and physical reprocessability (7.75% elongation retention after initial reprocessing). Additional functionalities include pressure-sensitive adhesion (interfacial toughness: 70.06 J/m^2^ on glass), thermally triggered shape memory behavior (fixed at −25 °C with reversible deformation/recovery at ambient conditions), and notable biodegradability (13.25% mass loss after 45-day soil burial). Molecular simulations reveal that the unique structure of the WMFAEE monomer enables a dual mechanism of autonomous self-healing at room temperature without external stimuli: chain diffusion and entanglement-driven gap closure, followed by hydrogen bond-mediated network reorganization. Furthermore, the synergy between monomer chain diffusion/entanglement and dynamic hydrogen bond reorganization allows the WMFAEE-HPA system to achieve a balance of multifunctional integration. Moreover, the integration of these multifunctional attributes highlights the potential of this WCO-derived photocurable elastomer for various possible 3D printing applications, such as flexible electronics, adaptive robotics, environmentally benign adhesives, and so on. It also establishes a paradigm for converting low-cost biowastes into high-performance smart materials through precision molecular engineering.

## 1. Introduction

The valorization of waste cooking oil (WCO) offers a sustainable pathway to repurpose a pervasive waste stream into high-value products, mitigating environmental contamination and health risks associated with its improper disposal [1,2]. Leveraging compositional parallels between WCO and vegetable oils, extensive research over two decades has focused on its potential as a renewable feedstock for oleochemical derivatives [3]. While biodiesel production remains the most industrialized application [4,5,6], WCO has also been utilized in synthesizing biolubricants [7,8,9], bioplasticizers [10,11], alkyd resins [12], detergents [13], and surfactants [14]. However, these conventional approaches face economic constraints, with profit margins typically below 20% [15], compounded by fragmented collection systems and regionally inconsistent infrastructure [16]. The heterogeneous nature of WCO, characterized by impurities, variable moisture content, and complex pretreatment requirements, further undermines product consistency and market competitiveness [17]. Consequently, large-scale WCO recycling via traditional routes demands substantial policy support and capital investment, limiting feasibility to developed economies [18,19]. To enable universal adoption, particularly in resource-limited regions, innovative strategies must prioritize high-value applications featuring simplified processing, low operational costs, and tolerance to feedstock variability [20]. Such advancements align with circular economy principles while addressing the dual challenges of waste management and sustainable material development.

In our prior investigations, WCO has been strategically repurposed into high-value-added products (such as solid alcohol [21], wax [22], photocurable 3D printing materials [23,24,25], and coatings [26]) through tailored chemical modifications leveraging its unsaturation degree. Among these, photocurable 3D printing materials exhibit exceptional economic potential, with value-added margins exceeding 100%, positioning them as a premium route for WCO upcycling. Conventional strategies involve either blending epoxidized WCO (E-WCO) as a plasticizer into photoresins to enhance mechanical properties [23] or synthesizing unsaturated ester-grafted photocurable monomers [24]. Notably, converting WCO into photocurable shape-memory polymers (PSMPs) unlocks advanced dynamic deformatable 3D printing applications (including biomedical scaffolds, smart devices, and self-adaptive systems [27,28,29]) surpassing the static limitations of traditional 3D printing materials. Compared to petroleum-derived PSMPs, WCO-based variants promise reduced costs and environmental footprints.

Building on this foundation, we previously developed a mild “epoxidation-esterification-blending” strategy to synthesize multifunctional 3D-printable resins. Epoxidation of WCO yields E-WCO, purifying the feedstock while introducing reactive epoxy groups. Subsequent ring-opening esterification with methacrylic acid (MAA) generates epoxy waste oil methacrylate (EWOMA), a photocurable monomer. By blending EWOMA with secondary monomers (e.g., TEGDMA or PHEA/MAA), PSMPs with tunable functionalities were achieved: rigid plastics (TEGDMA-based) [24] or flexible adhesives (PHEA/MAA-based) exhibiting 230% elongation at break [25]. However, EWOMA inherits WCO’s tripartite branched architecture, leading to densely cross-linked networks that intrinsically restrict chain mobility. This structural constraint imposes a mechanical ceiling (<300% elongation) and impedes the integration of advanced functionalities such as autonomous self-healing—critical limitations for high-performance elastomers.

Recent advancements in polymer science have prioritized the development of self-healing materials to extend the operational lifespan and reliability of polymeric systems. Self-healing mechanisms are broadly categorized into extrinsic and intrinsic systems [30]. Extrinsic systems employ embedded microcapsules or vascular networks to release healing agents (e.g., linseed oil [31] or WCO-filled alginate microcapsules [32]) upon damage, enabling rapid crack sealing with efficiencies exceeding 90%. However, their dependency on finite healing reservoirs restricts multiple repair cycles [33], limiting long-term utility. In contrast, intrinsic self-healing leverages reversible covalent or non-covalent interactions, such as Diels–Alder reactions [34], disulfide bonds [35], or supramolecular forces (metal–ligand coordination [36], hydrogen bonding [37], π–π stacking [38]), and can achieve unlimited, autonomous repair without external agents [39].

Among these, supramolecular interactions offer distinct advantages by decoupling healing functionality from the polymer backbone, enabling room-temperature recovery while preserving mechanical integrity [40,41]. Despite these innovations, the integration of intrinsic self-healing into waste-derived biopolymers remains largely unexplored. As an abundant and low-cost feedstock, WCO presents an ideal platform for sustainable elastomers due to its inherent triglyceride structure and modifiable unsaturation. Prior efforts focused on extrinsic WCO utilization, e.g., microencapsulation for asphalt repair [32], yet no studies have reported intrinsic self-healing systems derived directly from WCO.

To overcome the performance limitations imposed by WCO’s tripartite branched architecture, we engineered a molecular topology-driven strategy to synthesize a photocurable elastomer resin that integrates autonomous self-healing, high flexibility, multifunctionality, and 3D printability. This approach begins with a sequential transesterification, epoxidation, and ring-opening esterification process, which transforms the branched triglyceride structure of WCO into a linear photocurable monomer, i.e., WCO-based methacrylate fatty acid ethyl ester (WMFAEE) (Figure 1). The molecular architecture of the synthesized WMFAEE monomer integrated three functional structural units: (1) Photocurable module: Methacrylate groups, introduced via ring-opening esterification, served as photocurable moieties. Upon UV irradiation, these groups underwent radical copolymerization with unsaturated comonomers, covalently anchoring hydrogen-bonding motifs and flexible alkyl chains into the polymer backbone; (2) Hydrogen bonding module: Hydroxyl groups, generated from epoxy ring-opening reactions, acted as dynamic hydrogen-bonding sites, enabling autonomous self-healing through reversible bond dissociation and reconfiguration. Additionally, terminal ethoxy ester groups on the flexible chains participated as hydrogen bond acceptors, further reinforcing the dynamic network; (3) Flexible chain module: Long alkyl chains inherited from the native triglyceride structure facilitated rapid chain diffusion and entanglement. This viscoelastic behavior synergized with dynamic hydrogen bonding during the initial healing phase, promoting efficient crack closure via large-scale chain mobility and localized interfacial adhesion.

By copolymerizing WMFAEE with hydroxypropyl acrylate (HPA), we embedded these dual mechanisms, i.e., hydrogen bonding and entanglement of long alkyl chains, into the polymer backbone, creating a sustainable photocurable network that unifies advanced functionalities, including room-temperature self-healing, exceptional elasticity, pressure-sensitive adhesion, thermally triggered shape memory, weldability, reprocessability, and biodegradability. The WMFAEE-HPA elastomer represents the first multifunctional 3D-printable material with self-healing capability derived from WCO, surpassing the mechanical and functional constraints of prior systems like EWOMA. Unlike petroleum-based elastomers, this material leverages WCO’s inherent molecular versatility to achieve performance parity while eliminating reliance on non-renewable feedstocks. Its compatibility with high-resolution 3D printing enables various possible applications, such as flexible electronics, adaptive robotics, environmentally benign adhesives, eco-friendly smart packaging, wearable smart devices, foldable structures, personalized stickers, toys, and decorative items.

## 2. Result and Discussion

### 2.1. Spectroscopic Analysis of WMFAEE Monomer

The structural evolution of WCO into WMFAEE via the sequential transesterification-epoxidation-ring-opening esterification strategy was unequivocally validated by ^1^H NMR spectroscopy. As illustrated in Figure 2a, the ^1^H NMR spectrum of raw WCO (primarily triglycerides) exhibited two split quadruplets at *δ* 4.1–4.4 ppm, corresponding to the methylene protons (−CH_2_−O−CO−R) of the glycerol backbone, and a triplet at *δ* 5.0–5.1 ppm, attributed to the methine proton (−CH−O−CO−R) of the glycerol moiety. Following transesterification, these glycerol-related signals completely disappeared in the spectrum of WCO-based fatty acid ethyl ester (WFAEE). Instead, a distinct quadruplet emerged at *δ* 4.1–4.2 ppm, assigned to the methylene protons (−OCH_2_CH_3_) of the terminal ethoxy group in WFAEE, confirming the cleavage of the branched triglyceride architecture into linear ethyl esters.

The unsaturated double bonds in WCO and WFAEE, initially reflected by resonance peaks at *δ* 5.2–5.3 ppm, were entirely absent in the spectrum of WCO-based epoxidized fatty acid ethyl ester (WEFAEE). Concurrently, new characteristic proton signals emerged at *δ* 2.8–3.2 ppm, corresponding to the epoxy group (−CH(O)CH−), verifying the conversion of double bonds into epoxide functionalities during epoxidation.

Subsequent ring-opening esterification of WEFAEE with methacrylic acid (MAA) led to the near-complete disappearance of epoxy proton signals (*δ* 2.8–3.2 ppm). Simultaneously, two doublets appeared at *δ* 5.5–6.4 ppm, characteristic of the terminal =CH_2_ protons in the methacrylate groups. These spectral transitions confirmed the exhaustive consumption of epoxy groups and the successful grafting of photocurable methacrylate moieties onto the WEFAEE backbone, yielding the linear WMFAEE monomer.

The IR spectra (Figure 2b) further corroborated these structural transformations. After transesterification, the asymmetric stretching vibration of C−O−C bonds, initially observed at 1234 cm^−1^, shifted slightly to a higher wavenumber (1239 cm^−1^), confirming the conversion of triglycerides in WCO into WFAEE. Subsequent epoxidation eliminated the bending vibrations of −CH= units (825 cm^−1^). Simultaneously, characteristic epoxy absorption peaks emerged at 910 cm^−1^ and 850 cm^−1^, validating the formation of WEFAEE. Following ring-opening esterification, the epoxy absorption peaks of WEFAEE disappeared entirely, while new absorption bands corresponding to methacrylate groups appeared at 1630 cm^−1^ (C=C stretching) and 943 cm^−1^ (out-of-plane bending of =CH_2_). These spectral changes confirmed the successful grafting of MAA onto WEFAEE, yielding the linear photocurable monomer WMFAEE.

### 2.2. Spectroscopic Analysis of WCO-Based Photocurable Elastomer

The spectroscopic characterization of the optimized A4 sample (WMFAEE-to-HPA mass ratio of 2:3) was conducted to elucidate structural changes before and after photopolymerization. As shown in Figure 3a, the liquid WMFAEE-HPA elastomer exhibited distinct absorption peaks corresponding to unsaturated bonds, including the C=C stretching vibration (1605 cm^−1^), in-plane bending vibrations of =CH_2_ (1461 cm^−1^), out-of-plane bending vibrations of =CH_2_ (947 cm^−1^), in-plane bending vibrations of −CH= (1300 cm^−1^), and out-of-plane bending vibrations of −CH= (815 cm^−1^). These peaks significantly diminished or disappeared after photopolymerization, confirming the radical polymerization of C=C double bonds as the primary curing mechanism. The calculated double-bond conversion rate reached 82.7%, indicating that a substantial proportion of the WMFAEE and HPA monomers actively participated in the copolymerization process.

Additionally, the absorption peak at 3446 cm^−1^ was attributed to hydroxyl group stretching vibrations, while the peak at 1051 cm^−1^ corresponded to C−O bond stretching in hydroxylated moieties. These observations indicated abundant hydroxyl groups within the WMFAEE-HPA elastomer matrix. The hydroxyl-rich surface characteristics were further corroborated by X-ray photoelectron spectroscopy (XPS). The full XPS spectrum (Figure 3b) revealed predominant carbon (C) and oxygen (O) elements on the elastomer surface. High-resolution C1s analysis (Figure 3c) resolved four distinct peaks: C−C bonds (285.02 eV) from the polymer backbone, C−O bonds (286.04 eV) associated with hydroxyl groups, C=O bonds (286.80 eV), and ester-linked C−O bonds (289.26 eV). The O1s spectrum (Figure 3d) further confirmed the presence of carboxylate ester C=O bonds (531.95 eV), hydroxyl C−O bonds (532.75 eV), and O−H groups (533.35 eV). These results collectively demonstrated the successful integration of hydroxyl functionalities into the elastomer network, consistent with its self-healing and adhesive properties.

### 2.3. Thermal Analysis

Thermogravimetric analysis (TGA) of the 3D-printed WMFAEE-HPA elastomers and pure HPA (A1 sample) was conducted to evaluate their thermal stability (Figure 4a and Appendix A). The pure HPA exhibited excellent thermal stability, with an onset decomposition temperature of 345 °C. In contrast, the introduction of WMFAEE significantly reduced the thermal stability of the copolymerized system, and the decomposition temperature progressively decreased with increasing WMFAEE content. This trend corroborated the successful copolymerization of WMFAEE and HPA, as the incorporation of thermally labile WCO-based polymeric segments inherently lowered the overall thermal resistance. Notably, even the least thermally stable formulation (A6 sample) retained stability up to 191 °C, which remains sufficient for practical applications under standard conditions.

Differential scanning calorimetry (DSC) further revealed distinct glass transition behaviors within the temperature range of −25 °C to 25 °C (Figure 4b and Appendix A). The pure HPA sample displayed the highest glass transition temperature (T_g_) at 19.3 °C, whereas copolymerization with WMFAEE drastically reduced the T_g_ to sub-zero values. A progressive decline in T_g_ was observed with increasing WMFAEE content, indicating enhanced molecular chain mobility due to the plasticizing effect of WMFAEE. This reduction in T_g_ weakened intermolecular interactions within the copolymer network, thereby increasing conformational freedom of the polymer chains. The suppressed T_g_ (<0 °C for WMFAEE-HPA systems) positioned the material near its glass transition at ambient temperatures (0–40 °C), which critically underpinned its multifunctional attributes, including room-temperature self-healing, pressure-sensitive adhesion, and thermally triggered shape memory behavior.

### 2.4. Three-Dimensional Printing Behavior

To validate the applicability of the WCO-based elastomer in photocurable 4D printing, the linear regression curve of lnE versus C_d_ for the A4 sample was analyzed, yielding a penetration depth (D_p_) of 0.224 mm and a critical exposure energy (E_c_) of 55.44 mJ/cm^2^ (Figure 5a). The D_p_ parameter reflects the material’s sensitivity to variations in light intensity and exposure time. A lower D_p_ (<0.3 mm) enhances tolerance to process fluctuations (for instance, the light output deviation of low-cost LCD 3D printers could readily exceed ±10% in practical applications), ensuring consistent layer-by-layer curing and dimensional fidelity, which was important for high-precision applications requiring sub-50 μm features. In contrast, materials with higher D_p_ (>1.0 mm), such as some soybean oil-based resins (e.g., AESO [42]), might exhibit substantial Z-axis deviations (>5% error at 50 μm layer thickness) due to uncontrolled light penetration, rendering them unsuitable for intricate geometries (Table 1). While higher D_p_ may theoretically enable faster printing via thicker layers, this advantage is offset by the need for stringent light-source calibration to mitigate over-curing artifacts, as demonstrated in prior studies [43,44].

Notably, the A4 elastomer’s D_p_ (0.224 mm) was lower than some commercial resin (D_p_ = 0.314 mm [45]), achieving enhanced compatibility with high-resolution printing. Practical evaluations confirmed that the A4 resin achieved smooth surfaces and high dimensional fidelity. At a layer thickness of 50 μm, the printed specimen exhibited minimal Z-axis deviations of 0.24% (Figure 5b), while in-plane deviations were controlled to approximately 0.25% (Figure 5c). These results demonstrated the material’s robustness in maintaining geometric precision under standard printing conditions, further supporting its suitability for advanced 3D printing applications.

### 2.5. Mechanical Properties

As shown in Figure 6a, the mechanical performance of the WCO-derived elastomer was critically dependent on the synergistic copolymerization of WMFAEE and HPA. Pure WMFAEE monomer lacked sufficient crosslinkable double bonds for standalone photocuring, but copolymerization with HPA yielded a 3D-printable photocurable elastomer with enhanced flexibility and elasticity. The homopolymer of pure HPA (A1 sample) exhibited moderate flexibility at room temperature, achieving a tensile strength of 0.957 MPa and an elongation at break of 382.48%. However, the incorporation of WMFAEE significantly altered the mechanical behavior. As the WMFAEE content increased, the elongation at break initially rose and then declined, reaching an optimal balance at a WMFAEE-to-HPA mass ratio of 2:3 (A4 sample). This formulation demonstrated a tensile strength of 0.967 MPa and an elongation at break of 645.09%, nearly 1.7 times that of pure HPA. Notably, substituting WMFAEE with the branched epoxy waste oil methacrylate (EWOMA) failed to enhance mechanical performance. The EWOMA-HPA blend (A7 sample, 2:3 mass ratio) exhibited inferior properties, with a tensile strength of 0.63 MPa and elongation of 98.5%, underperforming even pure HPA.

The disparity stems from distinct molecular architectures. As illustrated in Figure 6b, pure HPA forms a linear polymer with tightly packed chains stabilized by interchain hydrogen bonding, which restricts further flexibility enhancement. In contrast, the WMFAEE-HPA copolymer introduces flexible long alkyl branches from WMFAEE, imparting a pronounced plasticizing effect that expands the conformational freedom of polymer chains. This structural loosening facilitates large elastic deformations, while hydroxyl groups on WMFAEE synergistically engage in hydrogen bonding with adjacent HPA chains, augmenting mechanical integrity. This dual mechanism (alkyl chain mobility enhancement and dynamic hydrogen bonding) enables unprecedented elasticity without compromising strength, which was unattainable in cross-linked EWOMA-based systems.

As benchmarked in Table 2, the WMFAEE-HPA elastomer exhibits superior flexibility relative to both bio-based and petroleum-derived counterparts. Its elongation-at-break (645.09%) surpasses that of the previously reported EWOMA-PHEA-MAA resin [25] by a factor of 2.8, underscoring the critical role of WMFAEE’s linear topology in mitigating the rigidity inherent to branched architectures. Furthermore, when compared to petroleum-based photocurable elastomers (such as the methyl acrylate (MA)/n-butyl acrylate (BA) copolymer [52]), the WMFAEE-HPA system demonstrates a 12% increase in elongation, achieving this enhancement using renewable feedstocks. This performance parity, coupled with its biodegradability and self-healing capability, positions WMFAEE-HPA as a sustainable alternative that transcends the environmental and mechanical limitations of conventional flexible resins. The divergence arises from WMFAEE’s unique capacity to chain mobility (via alkyl branches) and dynamic bonding (via hydroxyl groups), a dual mechanism absent in both rigid bio-based systems and petrochemical analogs.

On the other hand, the mechanical properties of the WMFAEE-HPA elastomer exhibited pronounced temperature dependence. At room temperature (25 °C), the A4 sample demonstrated high flexibility and elasticity. However, as the temperature decreased, its tensile strength marginally increased, while the elongation at break dropped sharply. At −25 °C, the material transitioned into a rigid glassy state, characterized by a tensile strength of 1.20 MPa and an elongation at break of 10.36% (Figure 6c). This abrupt transition highlighted its potential as a shape-memory material, where the glassy state at −25 °C served as the fixed temporary phase, while the rubbery state at 25 °C enabled reversible deformation and recovery.

Cyclic tensile testing further elucidated the dynamic behavior of the A4 elastomer. Under varying strains (Figure 6d), hysteresis loops were observed in all five cycles, with hysteresis energy increasing proportionally to strain magnitude. This trend was linked to strain-induced disruption of hydrogen bonds within the network. At 100% strain (Figure 6e), the largest hysteresis loop occurred during the first cycle, diminishing in subsequent cycles. This phenomenon suggested that hydrogen bond reorganization lagged during initial deformation but became more efficient in later cycles, reducing energy dissipation.

The integration of 3D printability, self-healing, pressure-sensitive adhesion, and shape memory properties positioned the WMFAEE-HPA elastomer as a versatile candidate for flexible wearable devices and deformable personalized products. For instance, a 3D-printed dart fabricated from the A4 elastomer adhered to a wooden frame solely via its intrinsic pressure-sensitive adhesive properties. In this state, the dart underwent substantial elastic deformation and recovery cycles without auxiliary fixation, demonstrating good functionality under repeated mechanical stress (Figure 6f).

### 2.6. Self-Healing Properties

The multifunctional WMFAEE-HPA elastomer (A4 resin) demonstrated autonomous self-healing at room temperature, enabled by its dynamic hydrogen-bonded network and flexible alkyl chains. As shown in Figure 7a, a controlled scratch (25 μm width × 1 mm depth) was introduced on the A4 elastomer surface using a razor blade. Microscopic analysis demonstrated progressive scratch closure at room temperature (25 °C), achieving near-complete healing within 48 h. In stark contrast, pure HPA polymers (A1 sample), despite possessing hydrogen-bonding capability, exhibited negligible self-repair under identical conditions, underscoring the critical role of WMFAEE’s molecular architecture.

The pristine A4 elastomer exhibited a tensile strength of 0.967 MPa and an elongation at break of 645.09%. After 24 h of autonomous healing at ambient conditions, the healed sample retained 20.68% tensile strength (0.20 MPa) and 57.82% elongation (371.85%), demonstrating partial yet significant restoration of mechanical integrity (Figure 7b). As shown in Table 3, the self-healing efficiency of the WMFAEE-HPA elastomer occupies an intermediate position among reported systems: its elongation recovery (57.82%) surpasses epoxy resin composites reinforced with carbon fibers (53% elongation recovery) [58], while lagging behind commercial acrylic elastomers relying on molecular diffusion and entanglement (85% tensile strength recovery) [59]. However, WMFAEE-HPA could achieve self-healing performance using a much lower cost compared with the petroleum-based systems (e.g., commercial acrylic elastomers), while incorporating ancillary functionalities such as pressure-sensitive adhesion, shape memory, and biodegradability. These properties, combined with its renewable WCO-derived composition, position WMFAEE-HPA as a versatile, sustainable alternative for applications demanding both dynamic repair and environmental compatibility—e.g., reusable adhesives, adaptive soft robotics, and transient biomedical devices. Unlike high-efficiency disulfide bond-based systems (91.8% elongation recovery) [60], which often require toxic modifiers or lack biodegradability, WMFAEE-HPA’s design aligns with circular economy principles, offering a scalable paradigm for waste-to-functional-material conversion without compromising ecological safety.

To elucidate the self-healing mechanism of the WMFAEE-HPA system, equilibrium and non-equilibrium molecular dynamics (MD) simulations were conducted using LAMMPS-7Aug19 software to probe molecular-level repair dynamics. Molecular models of WMFAEE-HPA and pure HPA were constructed, and a Cu substrate was introduced to create a standardized repair interface. After deleting the Cu substrate and allowing interfacial polymer contact under periodic boundary conditions, hydrogen bond evolution at the interface was quantitatively monitored over time.

As shown in Figure 8a, hydrogen bonds at the interface increased rapidly during the initial healing phase for both systems. However, WMFAEE-HPA exhibited a significantly higher hydrogen bond density (1737 bonds at equilibrium) compared to pure HPA (1250 bonds), representing a 39% enhancement. This disparity underscores the critical role of WMFAEE-derived hydroxyl and ethoxy ester groups in forming a dynamic hydrogen-bonding network. Importantly, these dynamic hydrogen bonds acted as “temporary anchoring points” during repair, guiding chain segments toward the interface to fill voids while enabling rapid bond reconfiguration. Concurrently, mean square displacement (MSD) analysis (Figure 8b) revealed enhanced chain mobility in WMFAEE-HPA, with a diffusion coefficient of 4.81 × 10^−7^ cm^2^/s, nearly double that of pure HPA (2.73 × 10^−7^ cm^2^/s). This accelerated diffusion was attributed to the flexible long alkyl chains inherited from WMFAEE, which promoted large-scale chain rearrangement and entanglement through localized conformational adjustments. The synergy between dynamic hydrogen bonding and chain entanglement facilitated efficient crack closure without compromising bond reversibility.

The molecular simulation results of interfacial healing and mechanical integrity revealed that after 10 ns of simulated healing at 25 °C, the WMFAEE-HPA system exhibited no discernible interfacial voids, indicating near-complete self-recovery. In contrast, pure HPA retained distinct interfacial defects, as evidenced by persistent cracks (Figure 8c). Simulated triaxial tensile testing at a strain rate of 10^−4^ ns^−1^ further demonstrated the divergent repair outcomes: the healed interface of pure HPA fractured completely under stress, confirming its lack of practical self-healing capability. Conversely, the WMFAEE-HPA interface retained partial molecular chain connectivity despite localized microcracks, with the interfacial region remaining partially intact (Figure 8d). This mechanical robustness arose from the dynamic entanglement network formed by alkyl chains, which synergized with hydrogen bonds to distribute stress and resist crack propagation.

Cohesive energy calculations (Figure 8e) confirmed stronger intermolecular interactions in WMFAEE-HPA (213.7  kcal/mol) compared to HPA (156.7  kcal/mol). The enhanced cohesion stemmed from two cooperative mechanisms: (1) hydrogen-bond-driven interfacial adhesion and (2) entanglement-mediated energy dissipation enabled by the flexible alkyl chains. Unlike static, densely cross-linked networks, the dynamic entanglement in WMFAEE-HPA allowed frequent interchain contact and self-adjustment, balancing mechanical integrity with reparability. This enhanced cohesion, driven by hydrogen bonding and alkyl chain entanglement, also directly correlated with the material’s mechanical robustness and pressure-sensitive adhesion.

Combined with molecular simulation, the self-healing mechanism, illustrated in Figure 9, arises from a synergistic interplay between molecular chain diffusion/entanglement and dynamic hydrogen bond reorganization. In pure HPA systems, hydrogen bonds alone cannot bridge microcrack gaps exceeding their effective interaction range, rendering them ineffective for macroscopic repair. Conversely, the self-healing process of the WMFAEE-HPA system involves a hierarchical two-phase mechanism: (a) Chain diffusion and entanglement-driven proximity adjustment: Flexible alkyl chains in WMFAEE facilitate large-scale chain migration, mechanically bridging crack interfaces via entanglement. Simultaneously, terminal ethoxy ester groups function as hydrogen bond acceptors, synergizing with hydroxyl donors to accelerate interfacial hydrogen bonding. These groups further serve as “temporary anchoring points”, guiding chain segments toward the interfacial voids to facilitate rapid defect filling while enabling dynamic bond reconfiguration; (b) Hydrogen bond network reformation: Once the interfacial gap was narrowed to the range of hydrogen bonding interactions, dynamic hydrogen bonds re-establish across the interface, restoring structural integrity. This dual mechanism (chain diffusion and entanglement-enabled gap closure followed by hydrogen bond-mediated network reformation) ensures autonomous healing without external stimuli at room temperature. The absence of such a hierarchical process in pure HPA networks explains their limited repair capacity, highlighting WMFAEE’s indispensable role in achieving intrinsic self-healing functionality.

### 2.7. Welding and Physical Reprocessing Properties

The intrinsic self-healing capability of the WMFAEE-HPA elastomer enabled its fabrication via customized processing techniques, such as thermal welding, thereby broadening its applicability in personalized manufacturing applications. The weldability of the A4 elastomer was evaluated by overlapping two fractured specimens (5 mm overlap) and thermally annealing them at 80 °C for 12 h (Figure 10a). The welded sample retained 30.35% of the original tensile strength and 19.97% of the original elongation (Figure 10b). This moderate welding efficiency arose from hydrogen bond reformation and long alkyl chain interdiffusion at the interface, though incomplete molecular entanglement limited full mechanical recovery.

Similarly, fractured WMFAEE-HPA elastomer specimens could be rapidly reprocessed into intact forms via thermal compression molding. This reprocessability not only enhanced the operational flexibility of the material but also significantly increased its reusability value in post-consumer waste streams, aligning with circular economy principles. To validate this performance, the A4 elastomer was ground into fragments and hot-pressed at 180 °C under 4 MPa for 1 h to assess reprocessability (Figure 11a). After three reprocessing cycles, the material retained 19.65% tensile strength and 7.75% elongation post-first cycle, with progressive brittleness observed in subsequent cycles (Figure 11b). Mechanical degradation might be attributed to defect accumulation (e.g., voids, chain scission) during reprocessing. FTIR analysis (Figure 11c) confirmed that there were no obvious changes in the chemical groups, indicating reprocessing primarily induced physical rather than chemical alterations.

### 2.8. Pressure-Sensitive Adhesive Properties

The WCO-based elastomer exhibited multifunctionality as a pressure-sensitive adhesive (PSA), validated through dynamic mechanical analysis (DMA) and interfacial adhesion tests. As shown in Figure 12a,b, pure HPA displayed a storage modulus (G′) significantly higher than its loss modulus (G″) across the temperature range of 15–40 °C, with a temperature-insensitive loss factor (tan δ) below 1, indicative of dominant elastic behavior. In contrast, the A4 sample (WMFAEE-HPA) demonstrated balanced viscoelasticity at room temperature (15–40 °C), where G′ and G″ were nearly equivalent, yielding a tan δ close to 1. This near-unity tan δ indicated optimal equilibrium between molecular diffusion (viscosity) and energy dissipation (elasticity), a hallmark of practical PSA performance. Pure HPA, lacking long alkyl chains from WMFAEE, failed to achieve this balance and thus showed negligible PSA functionality.

The PSA performance of the WCO-based elastomer was modulated by HPA content. Interfacial adhesion toughness on steel initially increased with HPA concentration, peaking at 24.88 J/m^2^ for the A4 sample (Figure 12c), which also adhered to a Φ 3.97 mm steel ball in rolling ball tests and sustained a 500 g load for 270 min. Substrate-specific adhesion tests (Figure 12d) revealed the highest toughness on glass (76.06 J/m^2^), followed by aluminum (58.12 J/m^2^), wood (47.78 J/m^2^), PLA (28.04 J/m^2^), steel (24.88 J/m^2^), and PMMA (15.36 J/m^2^). Notably, adhesion to pigskin (simulating human skin) was minimal (3.48 J/m^2^), ensuring safe handling of 3D-printed products without unintended adhesion.

The PSA functionality aligned with practical applications. Three-dimensional-printed objects adhered vertically, obliquely, or suspended to glass surfaces (Figure 12e), enabling reversible elastic deformation. Light finger pressure triggered durable adhesion on smooth substrates like glass or steel. Additionally, the WCO-based elastomer could serve as a manually applicable PSA: liquid resin was coated onto polylactic acid (PLA) substrates, covered with a transparent release film to eliminate bubbles, and cured via 405 nm UV irradiation for 60 s (Figure 12f). The cured adhesive layer allowed easy bonding of PLA components to target surfaces upon film removal.

### 2.9. Shape Memory Properties

The shape memory behavior of the 3D-printed WMFAEE-HPA elastomer (A4 sample) was systematically investigated. As illustrated in Figure 13a, the printed cross-shaped structure was readily deformed at room temperature (25 °C). The temporary shape was effectively fixed by cooling the sample to −25 °C in a dry ice-ethanol-ethylene glycol solution. Subsequent reheating to 25 °C triggered rapid shape recovery, restoring the original geometry within minutes.

Quantitative shape memory cycle tests (Figure 13b,c) demonstrated exceptional performance. The material achieved a shape fixation ratio (R_f_) of 99.92% at −25 °C and a shape recovery ratio (R_r_) of 99.10% at 25 °C, with a maximum recovery rate (V_r_) of 2.2%/min under a nitrogen atmosphere. Even after five consecutive cycles, the elastomer retained high R_f_ (98.93%) and R_r_ (98.33%), confirming robust cyclic stability. These results underscored the material’s capacity for reversible, temperature-driven actuation, attributable to its dynamic hydrogen-bonded network and tailored glass transition behavior.

### 2.10. Biodegradability

The biodegradation behavior of the WMFAEE-HPA elastomer (A4 sample) and a commercial petroleum-based 3D printing resin was evaluated through a 45-day soil burial experiment. Weight loss measurements revealed distinct degradation profiles between the two materials. The WMFAEE-HPA elastomer exhibited rapid initial biodegradation, achieving 8.02% mass loss within 5 days, with cumulative degradation reaching 13.25% after 45 days. In contrast, the commercial resin displayed negligible biodegradation, with only 0.98% and 1.95% mass loss at the same intervals, underscoring its environmental persistence (Figure 14a).

To distinguish biodegradation from potential leaching of unreacted reagents, pre-degradation samples were immersed in deionized water for 72 h at 25 °C. No detectable soluble residues were observed via liquid chromatography, confirming thorough removal of unreacted monomers during synthesis and purification. Microstructural analysis via metallurgical microscopy further corroborated biodegradation mechanisms. Pristine A4 samples (Figure 14b) displayed smooth, homogeneous surfaces, whereas post-degradation specimens (Figure 14c) exhibited pronounced roughness, microporosity, and microbial colonization patterns, indicative of polymer matrix erosion by soil microorganisms.

Despite its biodegradability in soil, the WCO-derived elastomer demonstrated robust stability under ambient conditions. Specimens stored in air showed no significant mass loss over 45 days, confirming that degradation was specifically triggered by microbial activity in soil (Figure 14a). Thermal stability assessments (Figure 4a) revealed an initial decomposition temperature near 200 °C, suitable for routine applications below this threshold. The rapid yet controlled biodegradation of WMFAEE-HPA aligns with circular economy principles, offering a sustainable alternative to persistent petroleum-based resins. These results highlight its dual functionality: environmental compatibility in end-of-life scenarios and operational stability during use.

## 3. Materials and Methods

### 3.1. Materials

The waste cooking oil (WCO) was collected from the dining facilities of Guilin University of Technology (Guilin, China). The iodine value of the WCO, which quantifies the degree of unsaturation as the mass of iodine (in grams) absorbed by 100 g of oil, was determined to be 108.2 g I_2_/100 g through standard titration methods. Fatty acid composition analysis via gas chromatography-mass spectrometry (GC-MS, Appendix A) further confirmed that oleic acid (C18:1) dominated the lipid profile, constituting 46.1 wt% of the total fatty acids (Appendix A). The hydrogen peroxide (H_2_O_2_, analytical grade, 30 wt% in H_2_O) and sulfuric acid (98%) were purchased from Xiya Chemical Technology Co., Ltd. (Linyi, China). Other synthetic reagents, including sodium ethylate (20% *w*/*w* ethanol solution), anhydrous ethanol (99.9%), glacial acetic acid (99.5%), urea (99%), sodium bicarbonate (99.5%), methacrylic acid (MAA, 98%, stabilized with 250 ppm 4-methoxyphenol), triphenylphosphine (PPh3, 99%), hydroquinone (HQ, 99%), phenylbis (2,4,6-trimethylbenzoyl) phosphine oxide (Irgacure 819, 98%), p-dimethylaminobenzaldehyde (DMAB, 99%), and hydroxypropyl acrylate (HPA, 95%, stabilized with 200 ppm of 4-methoxyphenol), were all obtained from McLean Company (Shanghai, China). As a control sample, the commercial 3D printing photocurable resin was purchased from Anycubic Technology Co., Ltd. (Shenzhen, China).

### 3.2. Preparation of WMFAEE Monomer

The WMFAEE monomer was synthesized from WCO via a sequential three-step strategy: transesterification, epoxidation, and ring-opening esterification, as detailed below:

(1) Transesterification of WCO

The first step involved converting WCO triglycerides into linear fatty acid ethyl esters (WFAEE) to enhance molecular flexibility. WCO primarily comprises triglycerides, where three fatty acid chains are esterified to a glycerol backbone. Transesterification replaced the branched triglyceride structure with monochain ethyl esters.

At first, a mixture of 10 mL sodium ethoxide and 50 mL anhydrous ethanol was prepared. And then, 100 g of centrifuged WCO was added to a three-neck flask, and the sodium ethoxide solution was introduced at 90 °C under continuous stirring for 2 h. Post-reaction, the product was washed with 0.1 mol/L HCl until neutral, followed by two washes with saturated NaCl solution to remove impurities. Residual water was removed via vacuum evaporation at 80 °C, yielding WFAEE as a light orange liquid.

(2) Epoxidation of WFAEE

Epoxidation introduced epoxy groups into the unsaturated bonds of WFAEE to enhance reactivity for subsequent functionalization.

At first, peracetic acid was synthesized by mixing 160 g glacial acetic acid, 680 g 30% H_2_O_2_, 4 g H_2_SO_4_, and 4 g urea in a light-shielded container. The mixture was incubated at 40 °C for ≥12 h. And then, 1000 g of WFAEE was loaded into a 5 L jacketed glass reactor equipped with a mechanical stirrer and reflux condenser. The reactor was heated to 40 °C via circulating water. The preformed peracetic acid was slowly added to WFAEE over 2 h via a constant-pressure funnel. The temperature was raised to 70 °C, and the reaction proceeded for 3 h under stirring. After 10 h of phase separation, the aqueous layer was discarded. The oil layer was washed with 5% NaHCO_3_ and 60 °C deionized water until neutral. The product, WCO-based epoxidized fatty acid ethyl ester (WEFAEE), was obtained as a pale yellow, odorless liquid after vacuum evaporation at 70 °C for 1.5 h. The WEFAEE showed an epoxy value of 4.18, while its Lovibond color values improved from Y = 20.2, R = 3.9 (WCO) to Y = 0.3, R = 5.2 (WEFAEE) (Appendix A).

(3) Ring-opening esterification of WEFAEE

The final step introduced photocurable methacrylate groups and hydroxyl functionalities via ring-opening of epoxy groups with methacrylic acid (MAA).

At first, 100 g of WEFAEE and 0.1 g of hydroquinone (inhibitor) were dissolved in a three-neck flask at 90 °C. Meanwhile, a solution of 45 g MAA and 1 g triphenylphosphine (PPh_3_) catalyst was prepared at 60 °C and added dropwise to the WEFAEE mixture. The reaction proceeded at 100 °C for 4 h under vigorous stirring. The crude product was washed with 5% NaHCO_3_ and deionized water to neutrality. Residual water was removed via rotary evaporation, yielding WCO-based methacrylate fatty acid ethyl ester (WMFAEE) as a transparent brown liquid. The WMFAEE monomer contained methacrylate groups that enabled UV-induced crosslinking for 3D printing, as well as hydroxyl groups that facilitated hydrogen bonding, contributing to self-healing and adhesion properties.

### 3.3. Synthesis of Liquid WMFAEE-HPA Elastomer

The liquid WMFAEE-HPA elastomer was synthesized by blending WMFAEE with varying ratios of HPA in a 60 °C water bath under continuous stirring. Subsequently, 2 wt% of the total mass of WMFAEE and HPA was allocated for the addition of phenylbis(2,4,6-trimethylbenzoyl)phosphine oxide (Irgacure 819, photoinitiator) and 4-dimethylaminobenzaldehyde (DMAB, accelerator). These components were sequentially introduced into the mixture to ensure homogeneous dissolution. The reaction mixture was stirred until a uniform, transparent yellow solution formed, indicating complete solubilization of all solids. To prevent premature photopolymerization, the final product was stored in amber glass bottles at room temperature under light-shielded conditions. The synthesis formulations for the WMFAEE-HPA elastomer and control samples were systematically designed and are detailed in Table 4.

### 3.4. Molding of WCO-Based Elastomer

The obtained liquid WCO-based elastomer could be photocuring molded using two methods.

(1) Three-dimensional printing

A cost-efficient Photon Mono 2 LCD 3D printer (Anycubic, Shenzhen, China) was employed for PSA molding, facilitating the transformation of liquid WCO-based elastomer formulations into customized, multifunctional 3D-printed architectures. This system utilized a 405 nm ultraviolet light source with an irradiance of 3.6 mW/cm^2^ to drive photopolymerization during layer-by-layer deposition. The optimized printing parameters were as follows: a bottom exposure time of 50 s, a normal exposure time of 15 s, and a layer thickness of 50 μm. The final 3D-printed products were transparent, light yellow, and slightly adhesive solids. After thoroughly cleaning the surface of the products with ethanol and drying them in air, they were ready for performance testing or other applications.

(2) Traditional photocuring coating

In the application of PSA, the liquid WCO-based elastomer could also be evenly applied onto substrates (such as metal foil, paper, cotton fabric, or polylactic acid plastic), followed by the placement of a transparent plastic release film to remove air bubbles. A 405 nm UV lamp was then used to irradiate the resins from the side of the transparent release film for 60 s, curing the resins and producing products with PSA properties. During use, the release film could be removed, allowing the product to adhere to specific surfaces.

### 3.5. Characterization

The physicochemical and functional properties of the synthesized WCO-derived photocurable elastomer were systematically characterized using established analytical protocols outlined in our previous work [24,25].

(1) Chemical and physical characterization

The iodine value of WCO was quantified following GB/T 5532-2008 [62], while the epoxy value of WEFAEE was determined via GB/T 1677-2008 [63]. Fatty acid composition analysis was performed through gas chromatography-mass spectrometry (GC-MS, Agilent 7890-5979, Santa Clara, CA, USA) after methyl ester derivatization in accordance with GB 5009.168-2016 [64]. Optical properties of WCO, WFAEE, and WEFAEE were evaluated using a LABO-HUB WSL-2 Lovibond tintometer (Xinrui, Shanghai, China). Rheological measurements of liquid elastomer formulations were conducted with an NDJ-8s rotational viscometer (Jitai, Shanghai, China), while UV-Vis absorption profiles were acquired using a Shimadzu UV3100 spectrophotometer (Tokyo, Japan).

(2) Spectroscopic analysis

FT-IR spectra (400–4000 cm^−1^, 4 cm^−1^ resolution) were recorded on a Nicolet 6700 spectrometer (Thermo Fisher Scientific, Waltham, MA, USA) using KBr pelletized samples. Double bond conversion (DC) during photocuring was calculated via the following [65]:(1)$$DC=(1-\frac{\left(A_{a}^{C=C}/A_{a}^{C=O}\right)}{\left(A_{b}^{C=C}/A_{b}^{C=O}\right)})\times 100\%$$
where the pre-irradiation (A_b_) and post-irradiation (A_a_) absorbance ratios were derived from in-plane bending vibrations of −CH= group (1300 cm^−1^) and C=O stretching (1744 cm^−1^) peaks. The X-ray photoelectron spectroscopy (XPS) of the cured resin was carried out with an ESCALAB 250Xi X-ray photoelectron spectrometer (Thermofisher, Waltham, MA, USA) with an Al Kα X-ray as the stimulating source.

(3) Three-dimensional printing performance

The 3D printing performance of WCO-derived photocurable pressure-sensitive adhesives (elastomers) was evaluated through determination of penetration depth (D_p_) and critical exposure energy (E_c_). The absorption of irradiation light by liquid photoresins generally conforms to the Beer-Lambert law, whereby light energy exhibits negative exponential attenuation along the irradiation depth. When the UV exposure exceeds a specific threshold (E_c_), the photoresin undergoes a phase transition from liquid to solid state. The curing depth (C_d_) can be mathematically described by the following equation [42]:(2)$$C_{d}=D_{p}\times \ln\frac{E}{E_{c}}$$(3)$$C_{d}=D_{p}\ln E-D_{p}\ln E_{c}$$
where C_d_ represents curing depth (mm), D_p_ denotes penetration depth (mm), E_c_ signifies critical exposure energy (mJ/cm^2^), and E indicates incident exposure energy (mJ/cm^2^).

Experimental measurements were conducted using a Photon Mono 2 LCD 3D printer (Anycubic, Shenzhen, China) to cure WCO-derived photocurable elastomers and control samples under varying exposure durations. This process generated cured films with different thicknesses. Subsequent determination of curing depths and corresponding incident exposure energies enabled the construction of lnE-Cd plots according to Equation (3). Nonlinear fitting through Origin 7.5 software yielded Dp and Ec values for each sample.

The printing accuracy was assessed by examining the microstructure of 3D-printed specimens using a Leica MC170 HD metallurgical microscope (Leica Microsystems, Wetzlar, Germany). This microscopic analysis provided quantitative evaluation of the printed structures’ dimensional fidelity and feature resolution.

(4) Thermal behavior

Thermogravimetric analysis (TGA) was conducted on a NETZSCH TG 209 F1 Libra analyzer (Netzsch, Selb, Germany) under a 10 °C/min heating ramp (25 to 600 °C).

Dynamic mechanical analysis (DMA) employed a Mettler Toledo DMA861e analyzer (Mettler-Toledo, Greifensee, Switzerland) in shear mode (1 Hz, 3 °C/min, −50 to 150 °C).

DSC thermograms were acquired on a TA DSC 204/2920 (TA Instruments, New Castle, DE, USA) under optimized thermal profiles: −60 °C equilibration → heating to 150 °C (10 °C/min) → cooling (3 °C/min). The glass transition behavior of elastomers was studied using data from the first cooling and the second heating cycles.

(5) Mechanical evaluation

Mechanical properties were assessed using an AG-20I universal testing machine (Shimadzu, Tokyo, Japan) at 100 mm/min on GB/T 1040.2-2006 [66]-compliant 1BA dumbbell specimens (Appendix A), with data averaged across quadruplicate measurements.

(6) Testing of self-healing, welding, and physical reprocessing properties

Scratch healing behavior was microscopically analyzed using a DYP-702C transmitted-reflected polarizing microscope (Dianying, Shanghai, China). Efficiency of self-healing, welding, and physical reprocessing was quantitatively determined via tensile testing performed on an AG-20I universal testing machine (Shimadzu, Tokyo, Japan) at a crosshead speed of 100 mm/min.

(7) PSA properties

Adhesion performance was evaluated using HG-series testers (Huaguo, Dongguan, China): (1) Initial adhesion: using an HG-810 initial adhesion tester according to the China Standard GB/T 4852-2002 [67] with a slope angle of 30°; (2) Holding adhesion time: using an HG-5 tape holding tack tester according to the China Standard GB/T 4851-2014 [68] under an external load of 500 g; (3) 180° peel strength: employing an HG-860 peel strength tester according to the China Standard GB/T 2792-2014 [69] at a fixed peeling rate of 100 mm/min. The interfacial adhesion toughness (Γ, J/m^2^) was calculated as follows:(4)$$\Gamma =\frac{2F}{\omega }$$
where F is the plateau force during peeling and w is the width of the specimen. Each adhesion result represents the average measurement of four samples having the same composition.

(8) Shape memory behavior

The shape memory curves were tested using a TA Q800 dynamic mechanical analyzer (TA Instruments, New Castle, DE, USA) under a nitrogen atmosphere. The test specimens measured 30 mm × 5 mm × 1.2 mm and were analyzed in tensile mode with a frequency of 1 Hz. The testing procedure was as follows: The sample was stretched to 10% strain at a constant strain rate of 6%/min under a nitrogen atmosphere. Subsequently, the temperature was reduced to −25 °C at a cooling rate of 2 °C/min, followed by isothermal holding at −25 °C for 3 min. After removing the external load, the strain of the temporary shape was recorded. During the free recovery step, the temperature gradually increased to 25 °C at a rate of 2 °C/min. The sample was then held isothermally at 25 °C for 45 min to observe the free recovery behavior.

To quantify the shape memory behavior, the shape fixity ratio and the shape recovery ratio were calculated as follows:(5)$$R_{f}=\frac{\varepsilon_{f}}{\varepsilon_{m}}\times 100\%$$(6)$$R_{r}=\frac{\varepsilon_{f}-\varepsilon_{r}}{\varepsilon_{f}}\times 100\%$$
where ε_m_, ε_f_, and ε_r_ represent maximum, fixed, and residual strains, respectively.

(9) Biodegradation testing

The biodegradability test was evaluated through soil burial tests. Initially, square specimens measuring 10 × 10 × 2 mm were printed using a 3D printer and placed in containers filled with garden soil. The specimens were buried 50 mm below the soil surface. Multiple sets of samples (including control samples) were prepared. Subsequently, the containers were placed in a humidity chamber with controlled temperature conditions set at 25 °C and 30% relative humidity, maintained for a duration of 45 days. At specific intervals, samples were retrieved, cleaned, and vacuum-dried at 25 °C for 24 h. The weights of the samples were measured before (W_before_) and after (W_after_) the biodegradation test, and the biodegradation rate (R_b_, measured by the weight loss, %) was calculated using the following formula [70]:(7)$$R_{b}=\frac{W_{before}-W_{after}}{W_{before}}\times 100\%$$

The microstructures of the samples before and after biodegradation were analyzed using a Leica MC170 HD metallurgical microscope (Leica Microsystems, Wetzlar, Germany). Soluble residues from the leaching test were analyzed using an Ultimate 3000 UHPLC-Q Exactive liquid chromatography-mass spectrometry system (Thermo Fisher Scientific, Waltham, MA, USA).

(10) Molecular simulations

Equilibrium and non-equilibrium molecular dynamics simulations were carried out to investigate the self-healing ability of HPA and WMFAEE-HPA at the molecular level using LAMMPS-7Aug19 [71]. HPA and WMFAEE-HPA were modeled using GAFF2 force fields and RESP charges. All systems were annealed at 500 K and 400 K for 1 ns, respectively, and then relaxed to equilibrium state at room temperature for another under NPT ensemble. Cu was introduced at the bottom and top of the simulation box as a matrix. The simulated box was stacked in the z-axis in order to build a repair interface [72]. The snapshots of HPA and WMFAEE-HPA before self-repair are shown in Appendix A.

When the simulation was started, all Cu matrices were deleted, and polymers started to meet at the interface, where self-repair progress began under room temperature for as long as 10 ns. After self-healing, to evaluate each polymer’s property, we performed NEMD simulation under triaxial extension with extension rate ε = 10^−4^ ns^−1^ for 50 ps. The extension was assumed along the Z-axis, which was perpendicular to the repaired interface. All hydrogen bonds and cohesive energies were collected or calculated during the last 0.5 ns in equilibrium, and all snapshots were captured using OVITO-3.11.3 [73]. The cohesive energy (E_coh_), defined as the energy required to vaporize one mole of condensed-phase material by overcoming intermolecular interactions, was calculated using the following equation [72].(8)$$E_{coh}=\frac{\sum_{i=1}^{n}U_{i}-U_{condense}}{N}$$
where the U_i_ and U_condense_ represent the potential energies of an isolated molecular chain and the condensed phase in the equilibrium state, respectively. These energy values were derived from statistical averages computed over the last 0.5 ns of the simulation trajectory (Appendix A).

## 4. Conclusions

This study successfully synthesized a multifunctional 3D-printable photocurable elastomer derived from WCO through a sequential strategy of transesterification, epoxidation, and ring-opening esterification, yielding the linear photocurable monomer WMFAEE. By copolymerizing WMFAEE with HPA, the resultant elastomer exhibited exceptional mechanical flexibility (elongation at break: 645.09%), autonomous room-temperature self-healing (57.82% elongation recovery), weldability (19.97% elongation retention), physical reprocessability (7.75% elongation retention), and good PSA properties (interfacial toughness: 76.06 J/m^2^ on glass). Meanwhile, the material demonstrated thermally triggered shape memory behavior with 99.92% shape fixation at −25 °C and 99.10% recovery at 25 °C, alongside remarkable biodegradability (13.25% mass loss after 45-day soil burial).

The molecular simulation results demonstrated that the unique architecture of the WMFAEE monomer enabled a dual-mechanism healing process: (1) chain diffusion and entanglement-driven gap closure, followed by (2) hydrogen bond-mediated network reconfiguration. This hierarchical mechanism ensured autonomous self-healing at ambient temperature without external stimuli. The synergistic interplay between dynamic chain mobility (facilitated by flexible alkyl segments) and reversible hydrogen bonding (via hydroxyl and ethoxy ester motifs) allowed the WMFAEE-HPA system to achieve an unprecedented balance of multifunctional properties. These findings underscore the critical role of molecular topology design in integrating antagonistic properties (e.g., mechanical robustness and dynamic reparability) within a single material system, offering a sustainable paradigm for advanced elastomers in flexible electronics, adaptive robotics, and transient biomedical devices.

Compared to petroleum-based elastomers, the WCO-derived elastomer offered distinct advantages, including low cost, sustainable sourcing, and multifunctionality, while outperforming previous WCO-based systems in elongation and healing efficiency. Its 3D printability and tunable viscoelasticity further supported applications in many possible applications, such as flexible electronics, adaptive robotics, environmentally benign adhesives, eco-friendly smart packaging, wearable smart devices, biological scaffolds, foldable structures, personalized stickers, toys, and decorative items.

From a circular economy perspective, this work established a scalable pathway to valorize low-value biowaste into high-performance smart materials. The synthesis process featured mild reaction conditions, minimal secondary pollution, and high profitability, addressing both environmental and economic challenges associated with WCO disposal. By aligning with green chemistry principles, the developed elastomer not only mitigates reliance on non-renewable resources but also contributes to sustainable manufacturing practices. Future work will focus on optimizing reprocessability and scaling production for industrial adoption, further advancing the paradigm of waste-to-wealth conversion in materials science.

## Figures and Tables

**Figure 1 molecules-30-01824-f001:**
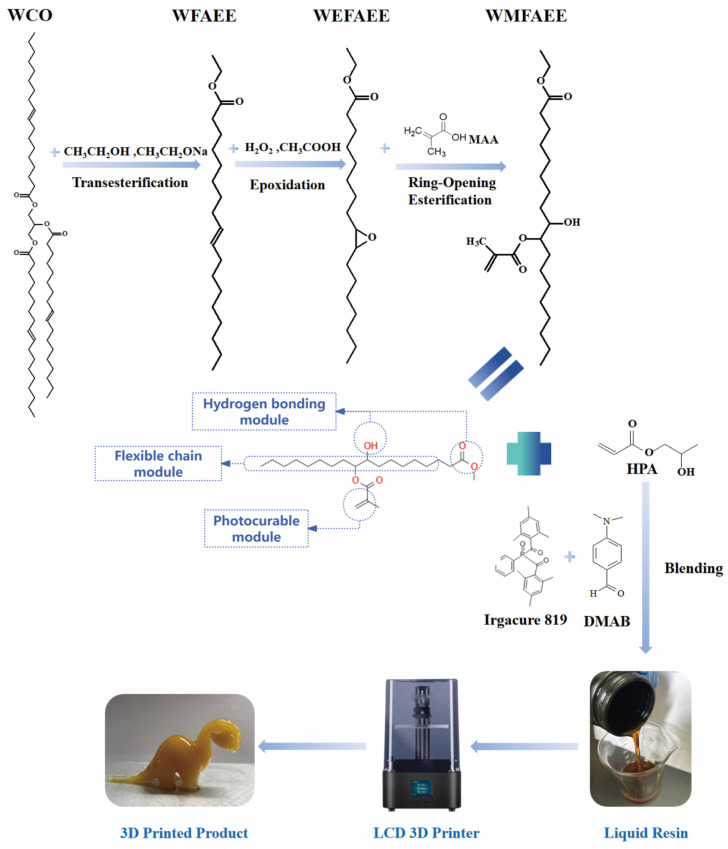
Schematic illustration of the preparation and 3D printing process of the WCO-based photocurable elastomer.

**Figure 2 molecules-30-01824-f002:**
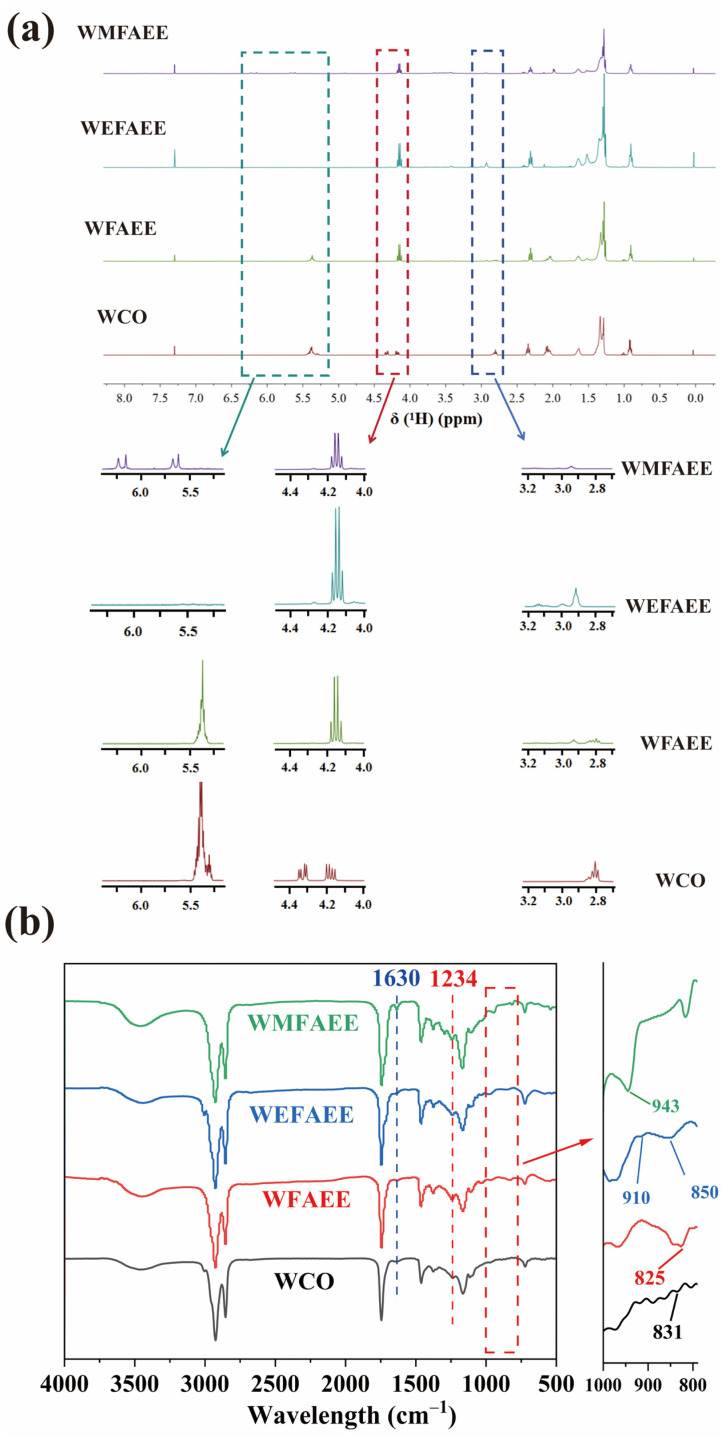
^1^H NMR (**a**) and IR spectra (**b**) of WMFAEE and its synthetic intermediates.

**Figure 3 molecules-30-01824-f003:**
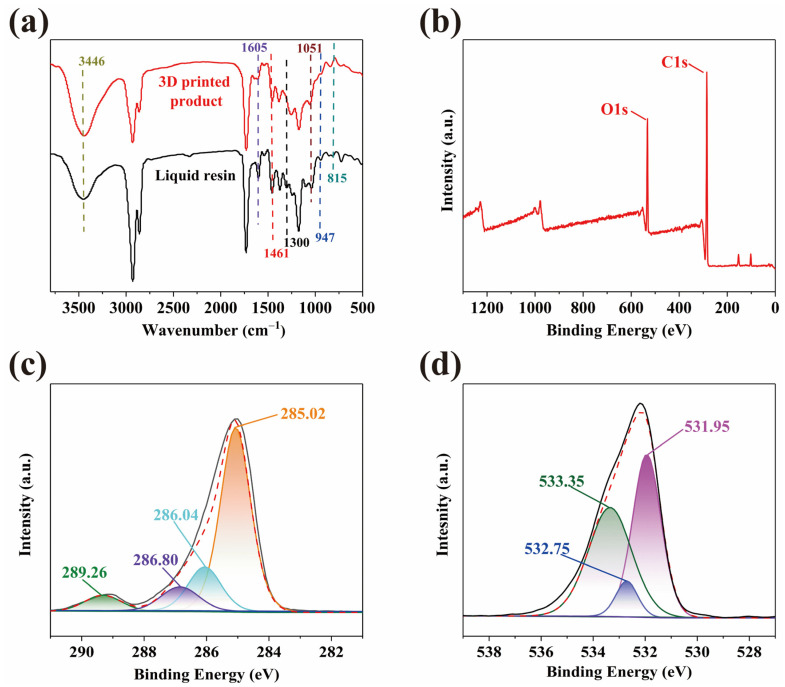
Spectroscopic analysis of WMFAEE-HPA elastomer (A4 sample): (**a**) IR spectra of 3D-printed product and liquid resin; (**b**–**d**) The full XPS (**b**), C1s (**c**), and O1s (**d**) high-resolution XPS spectra of the 3D-printed product.

**Figure 4 molecules-30-01824-f004:**
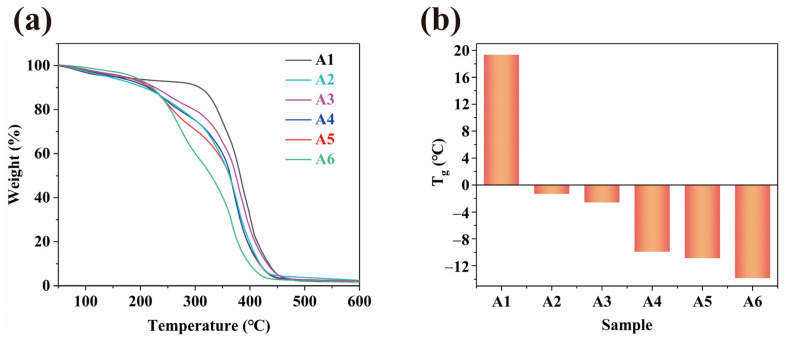
TGA curves (**a**) and T_g_ values (**b**) of of WMFAEE-HPA elastomers and pure HPA (A1 sample).

**Figure 5 molecules-30-01824-f005:**
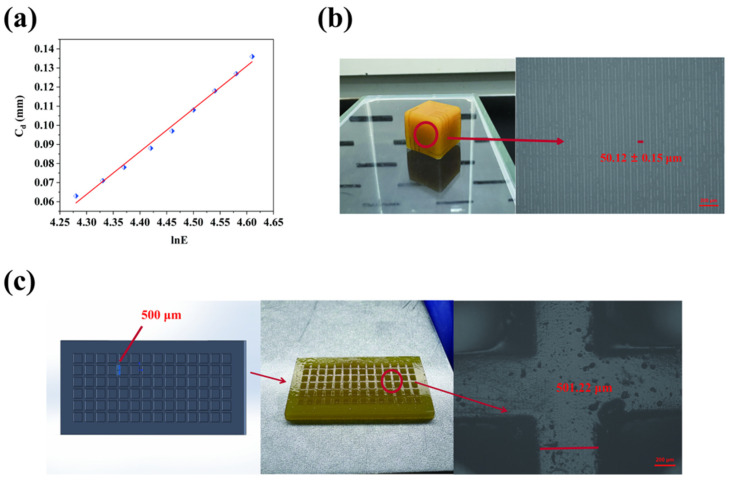
Three-dimensional printing behavior of WMFAEE-HPA elastomer (A4 sample): (**a**) Linear regression curve of lnE versus C_d_; (**b**) 3D-printed space dice specimen (**left**) and its 50× magnified metallographic micrograph (**right**) fabricated using A4 elastomer; (**c**) Computer-aided design model of lattice-structured cuboid (**left**), 3D-printed product using A4 elastomer (**middle**), and corresponding 50× magnified metallographic micrograph (**right**).

**Figure 6 molecules-30-01824-f006:**
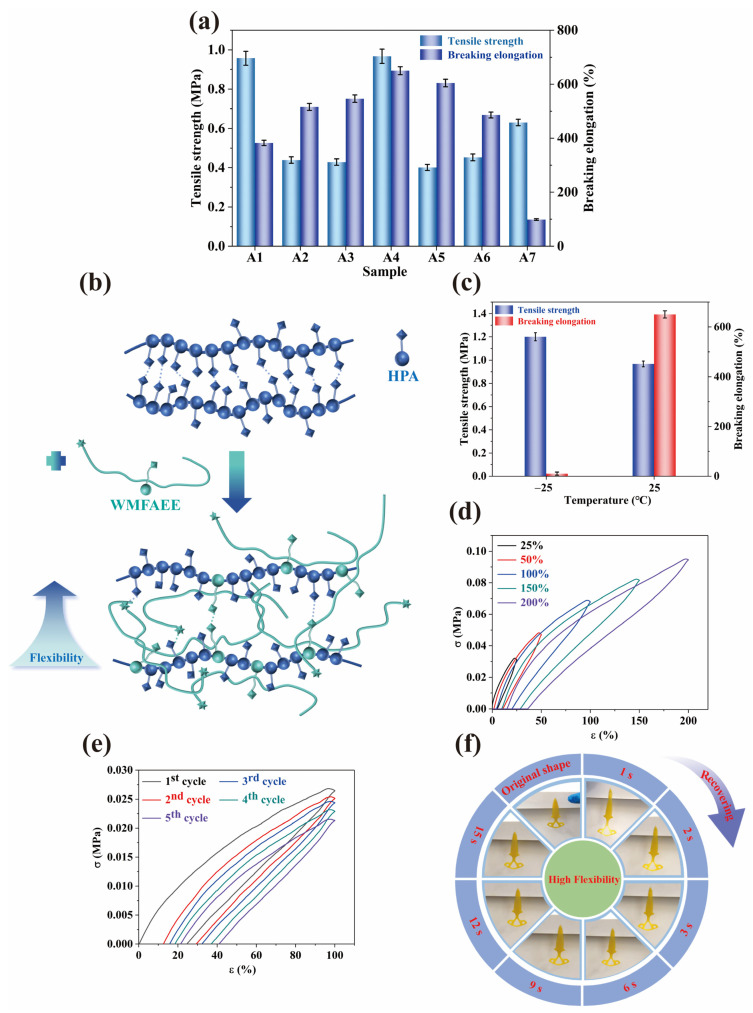
Mechanical properties of the WCO-derived elastomer: (**a**) Comparative mechanical performance of WMFAEE-HPA elastomers with varying HPA content; The pure HPA (A1) and EWOMA-HPA (A7) were both listed as control samples; (**b**) Schematic illustration of the structural differences between the curing products of WMFAEE-HPA elastomer and pure HPA. Spherical, cubic, and hexagonal-star icons were employed to schematically represent methacrylate, hydroxyl, and ethoxy ester functional groups, respectively. (**c**) Mechanical properties of the A4 elastomer at different temperatures; (**d**) Stress–strain curves from cyclic tensile testing (five cycles) of the A4 elastomer under varying strains; (**e**) Stress–strain profiles of the A4 elastomer during five-cycle tensile testing at 100% strain; (**f**) Elastic stretching deformation and recovery cycles of a 3D-printed dart made from the A4 elastomer, adhered to a wooden frame via the material’s intrinsic pressure-sensitive adhesive properties without auxiliary fixation.

**Figure 7 molecules-30-01824-f007:**
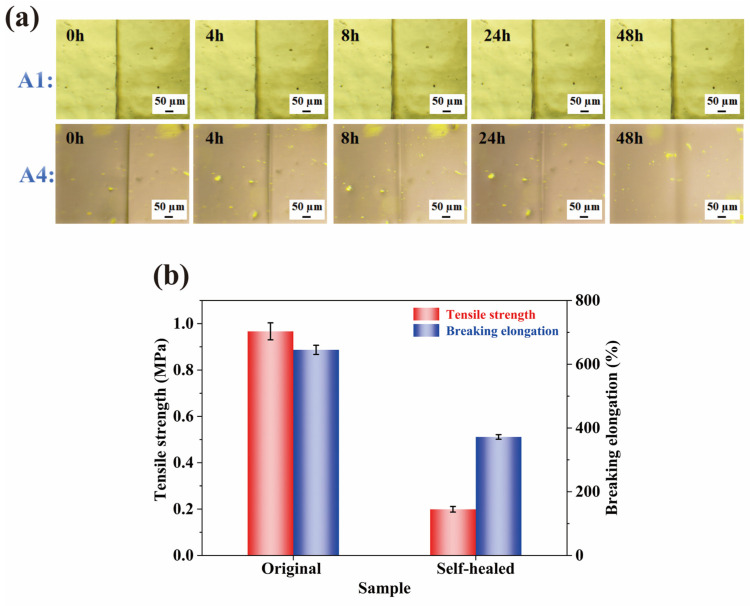
Self-healing performance of WMFAEE-HPA elastomer (A4 sample) at room temperature: (**a**) Scratch healing process of A4 elastomer vs. pure HPA (A1 sample); (**b**) Mechanical properties of A4 elastomer before and after 24 h self-healing.

**Figure 8 molecules-30-01824-f008:**
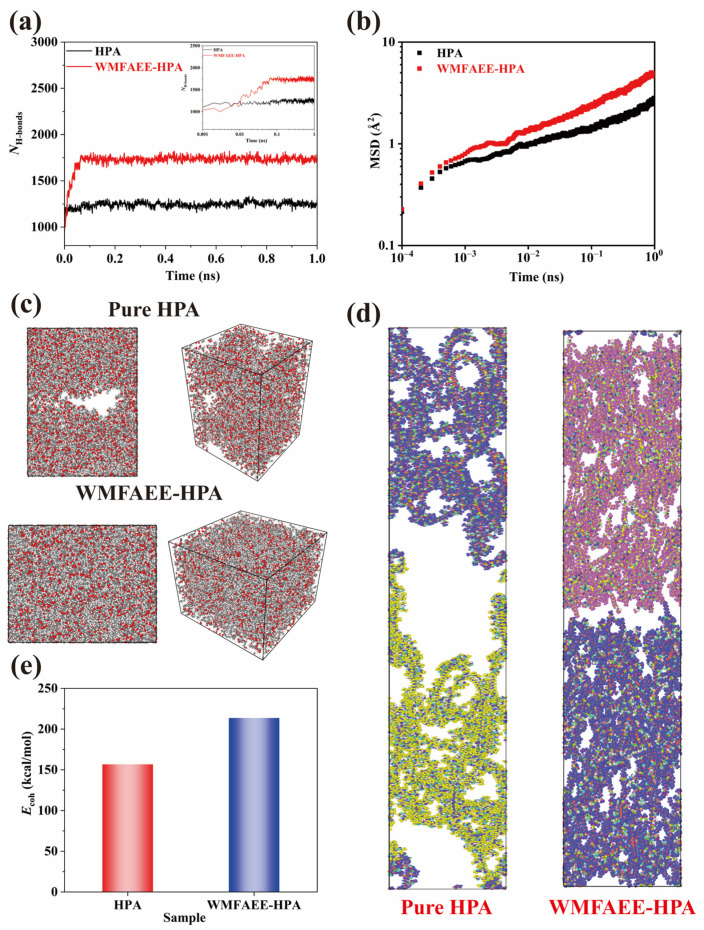
Molecular dynamics simulations of WMFAEE-HPA and pure HPA systems: (**a**) The number of hydrogen bonds as a function of time during self-healing, while the inset was semi-log plot; (**b**) The MSD values of HPA and WMFAEE-HPA during self-healing; (**c**) The snapshots of HPA and WMFAEE-HPA after self-healing for 10 ns; (**d**) The snapshots of self-healed HPA and WMFAEE-HPA after extension; (**e**) The comparison of cohesive energy of HPA and WMFAEE-HPA.

**Figure 9 molecules-30-01824-f009:**
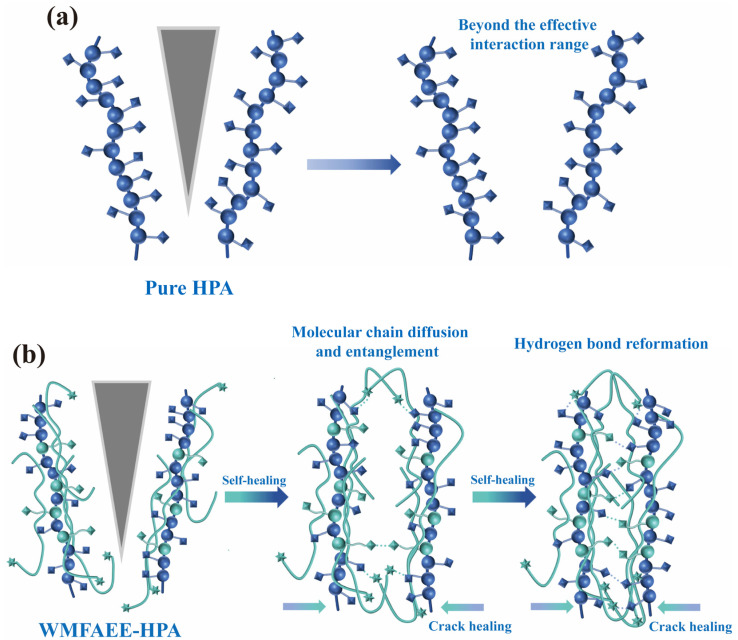
Schematic illustration of self-healing mechanism of WMFAEE-HPA elastomer; Spherical, cubic, and hexagonal-star icons were employed to schematically represent methacrylate, hydroxyl, and ethoxy ester functional groups, respectively.

**Figure 10 molecules-30-01824-f010:**
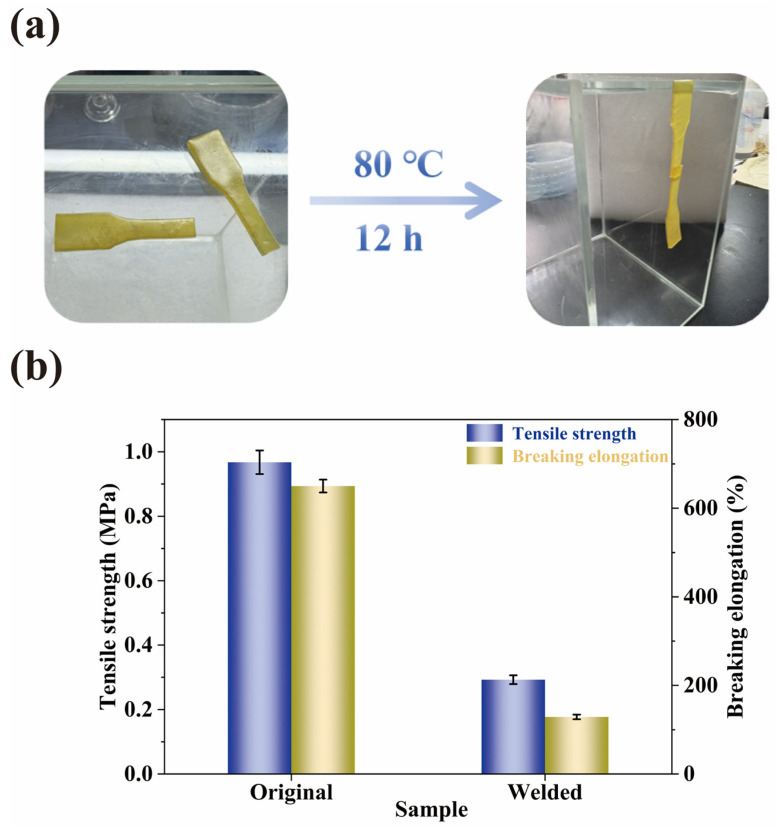
Welding performance of WMFAEE-HPA elastomer (A4 sample): (**a**) Photographs of welded tensile specimens before and after 12 h welding; (**b**) Mechanical properties of A4 elastomer before and after welding.

**Figure 11 molecules-30-01824-f011:**
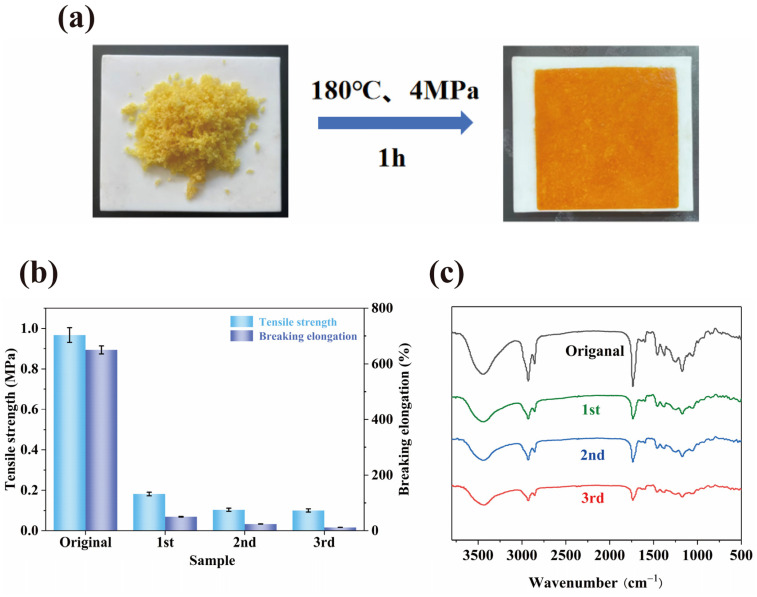
Physical reprocessability of WMFAEE-HPA elastomer (A4 sample): (**a**) Photographs of fragments and hot-pressed sheets after the reprocessing process; (**b**) Mechanical properties of A4 elastomer before and after different reprocessing cycles; (**c**) IR spectra of original and reprocessed samples.

**Figure 12 molecules-30-01824-f012:**
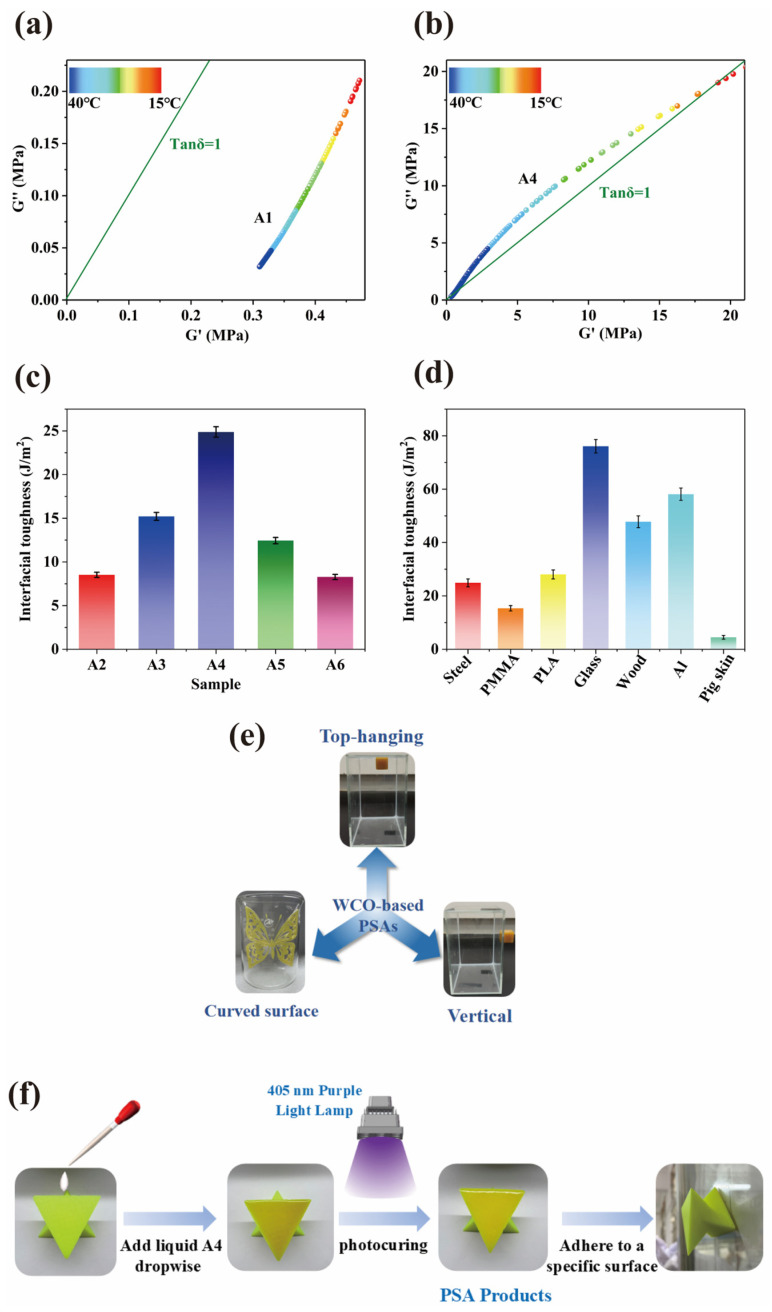
PSA properties of WMFAEE-HPA elastomer: (**a**,**b**) Viscoelastic curves of pure HPA (**b**) and A4 sample (**b**) at different temperatures ranging from 15 °C to 40 °C; (**c**) Interfacial adhesion toughness of WMFAEE-HPA elastomer on steel surface with different amounts of HPA; (**d**) Interfacial adhesion toughness of A4 elastomer on different adhesion substrates at room temperature; (**e**) Demonstration of PSA performance in 3D-printed A4 elastomer products: Left: Butterfly adhered to a curved glass surface; Upper: Dice suspended from an overhead substrate; Right: Dice vertically adhered to a flat glass substrate; (**f**) Manual PSA application and curing workflow for PLA-glass adhesion.

**Figure 13 molecules-30-01824-f013:**
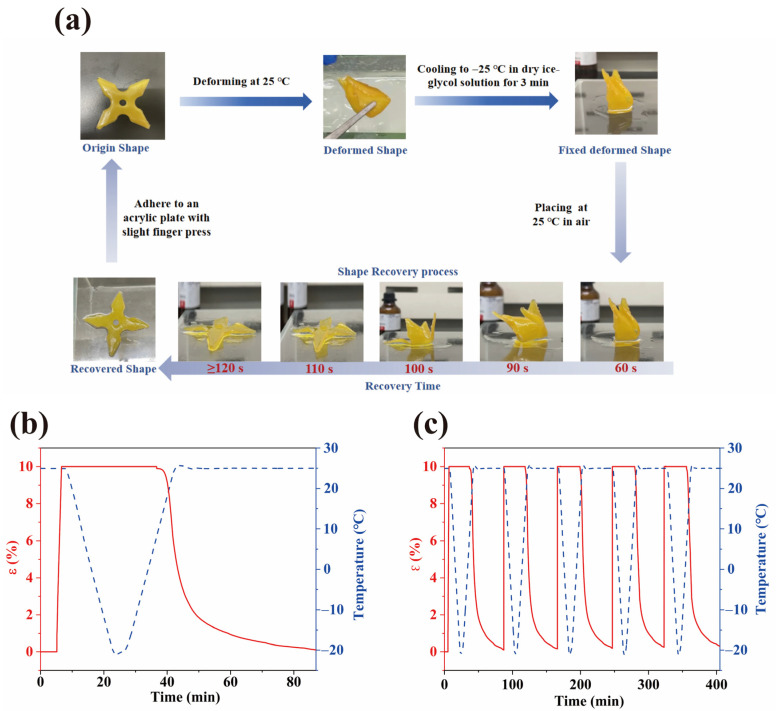
Shape memory behavior of the A4 elastomer: (**a**) Shape memory cycle demonstration of a 3D-printed cross-shaped structure; (**b**) Strain–temperature profile during a single shape memory cycle; (**c**) Fixation and recovery ratios over five consecutive cycles.

**Figure 14 molecules-30-01824-f014:**
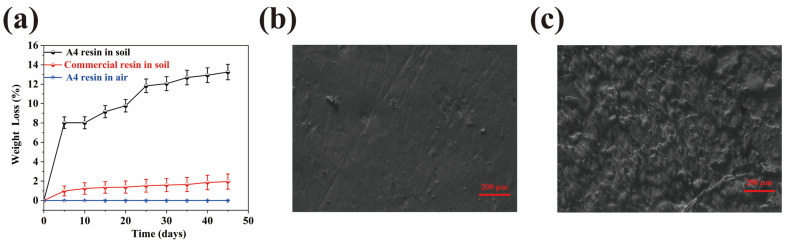
(**a**) Comparative biodegradation rates of the WMFAEE-HPA elastomer (A4 sample) and commercial 3D printing resin during soil burial; (**b**,**c**) Surface microstructures of A4 sample before (**b**) and after (**c**) 45-day soil burial experiment.

**Table 1 molecules-30-01824-t001:** Comparison of D_p_ and E_c_ values for some reported 3D printing materials.

Major Compositions	D_p_ (mm)	E_c_ (mJ/cm^2^)	Reference
WCO-based methacrylate fatty acid ethyl ester (WMFAEE) Hydroxypropyl acrylate (HPA)	0.224	55.44	This work
Acrylated epoxidized soybean oil (AESO)	1.318	77	[42]
Commercial resin	0.314	16.32	[45]
Bifunctional vanillin-vased vuilding vlock (It contains Priamine 1075 and MEV) Trifunctional crosslinker (It contains TREN and MEV) Isobornyl methacrylate (IBOMA) Phenylbis(2,4,6-trimethylbenzoyl) phosphine oxide (BAPO)	0.404	17.5	[46]
Methacrylated vanillin (MV) Glycerol dimethacrylate (GDM)	0.340	187	[47]
Choline chloride (ChCl) Hydroxyethyl methacrylate (HEMA) Tannic acid (TA)	0.192	8.70	[48]
Poly(dimethylsiloxane) (PDMS) PDMS macromolecular chains with methacryloxypopyl functional groups at the terminal ends (3DP-PDMS-E) PDMS copolymers containing methacryloxypropyl functional groups in the side chains (3DP-PDMS-S)	0.213	-	[49]
Acrylic acid (AA) ZnCl_2_ H_2_O Citric acid (CA)	0.24	-	[50]
Castor oil (CO) Isophorone diisocyanate (IPDI) 2-(tert-Butylamino) ethyl methacrylate (TBEM)	0.176	-	[51]

**Table 2 molecules-30-01824-t002:** Comparison of the mechanical performance of the WMFAEE-HPA elastomer with previously reported photocurable flexible resins.

Major Compositions	Elongation at Break (%)	Tensile Strength (MPa)	Reference
WCO-based methacrylate fatty acid ethyl ester (WMFAEE) Hydroxypropyl acrylate (HPA)	645.09	0.967	This work
Epoxy waste oil methacrylate (EWOMA) 2-phenoxyethyl acrylate (PHEA) Methacrylic acid (MAA)	230.1	0.48	[25]
Poly(tetrahydrofuran-based) polyurethane acrylate oligomer (PPTMGA) Isobornyl acrylate (IBOA)	414.3 ± 7.6	15.7 ± 0.7	[44]
Methyl Acrylate (MA) n-Butyl Acrylate (BA)	575.15	16.97	[52]
Diglycidyl ether of bisphenol A (DGEBA) 1,4-Cyclohexanedicarboxylic acid (CHDA) Sebacic acid (SA) 1,5,7-Triazabicyclo [4.4.0] dec -5-ene (TBD)	280	22	[53]
Epoxidized derivatives of natural rubber (ENR) TEMPO-oxidized cellulose nanocrystals (TOCNs)	750	5.8	[54]
Poly(2-hydroxyethyl methacrylate) (PHEMA)	130	25.4	[55]
Palm oil fatty acid—ethylacrylamid (POFA-EA) Acrylic acid (AA) ZnCl_2_	867	2.1	[56]
Vinyl-terminated polydimethylsiloxane (VPS-22000) Branched mercapto-functionalized polysiloxane (MPS) Precipitated silica (PSi)	1400	-	[57]

**Table 3 molecules-30-01824-t003:** Comparison of the self-healing performance of the WMFAEE-HPA elastomer with previously reported self-healing polymers.

Major Compositions	Self-Healing Efficiency Based on Elongation at Break	Self-Healing Efficiency Based on Tensile Strength	Possible Self-Healing Mechanism	Reference
WCO-based methacrylate fatty acid ethyl ester (WMFAEE) Hydroxypropyl acrylate (HPA)	57.82%	20.68%	Molecular chain diffusion and entanglement + hydrogen bonding	This work
Epoxy resin (Bisphenol F diglycidyl ether (DGEBF), Bisphenol A diglycidyl ether (DGEBA) and1,6-Hexanediol diglycidyl ether) Carbon fibre	53%	-	Thermally driven physical flow filling for crack healing	[58]
Commercial acrylic elastome		85%	Molecular chain diffusion + hydrogen bonding	[59]
Cystamine 2,2-Bis [4-(3,4-dicarboxyphenoxy)phenyl]propane dianhydride (BPADA) 2,2-Bis [4-(4-aminophenoxy)phenyl]hexafluoropropane (HFBAPP) Methyl-2-pyrrolidinone (NMP) N,N-dimethylacetamide (DMAc) Triethylamine (TEA)	91.8%	-	Dynamic disulfide bonds	[60]
Poly(tetramethylene ether glycol) (PTMEG) Isophorone diisocyanate (IPDI) 2,2-Bis(hydroxymethyl) propionic acid (DMPA) Chitosan	-	35%	Hydroxyl-driven dynamic exchange reaction	[61]

**Table 4 molecules-30-01824-t004:** Formulation for the synthesis of WMFAEE-HPA elastomer.

Sample	WMFAEE (g)	EWOMA (g)	HPA (g)	Mass Ratio of WMFAEE (or EWOMA) to HPA	819 (g)	DMAB (g)
A1 (pure HPA, control sample)	0	-	100	0:1	2	2
A2	50	-	150	1:3	4	4
A3	50	-	100	1:2	3	3
A4	100	-	150	2:3	5	5
A5	100	-	100	1:1	4	4
A6	150	-	100	3:2	5	5
A7 (EWOMA-HPA, control sample)	-	100	150	2:3	5	5

## Data Availability

No new data were created or analyzed in this study. Data sharing is not applicable to this article.

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
