# Peer review of "Multifunctional 3D-Printable Photocurable Elastomer with Self-Healing Capability Derived from Waste Cooking Oil"

_molecules, 2025, doi:10.3390/molecules30081824_

Round 1

Reviewer 1 Report

Comments and Suggestions for Authors

In this work, the authors produced the photo-curable monomers from waist cooking oil by chemical modification and made resins by their co-polymerization with HPA. The reviewer thinks this kind of green chemistry is important for our society, thus the reviewer basically agrees publication of this manuscript after some modifications.

1.
The reviewer thinks that the printing method shown in this paper should be called as 3D printing because the authors just simply printed 3D objects. If the authors want to call the technique as 4D printing, explain the reason why this technique can be called as 4D printing.

2.
The authors mention that the long alkyl chain promote interchain entanglement, but it is questionable. Typically, bulky side groups make the polymer thinker and more rigid (large Kuhn length), leading less entanglement. 

3.
The authors should show glass transition temperature of the materials because the shape memory, self-healing ability and pressure-sensitive adhesion are typically realized by using polymers near the glass transition.

4.
Fig.1: Ring-ping should be Ring-Opening

5. 
Weight loss in a soil would not mean (bio)degradation but may mean leaching of unreacted reagents from the material. Please check whether the leaching did not occur during the experiment.

Reviewer 2 Report

Comments and Suggestions for Authors

Summary

            Waste cooking oil (WCO) was processed by transesterification, epoxidation, followed by ring-opening esterification to yield a methacrylated fatty-ester. This was copolymerized with hydroxypropyl acrylate (HPA) with Irgacure 819 and dimethylaminobenzaldehyde via a 3-D printer, to yield a solid product that self-heals at room temperature and biodegrades over time. The solid product exhibited the best tensile strength and breaking elongation properties when a 2:3 ratio of methacrylated fatty-ester : HPA was copolymerized. The copolymer self-healed to near completion at room temperature over the course of 48 hours, 645% elongation until break, 76 J/m2 interfacial toughness on glass, and suffered a 13% mass loss over 45 days in soil. This is all an effort to make use of waste oils that are difficult to profit from.

General Conceptual Concerns

            The chemical analyses in this paper are very weak compared to the physical analyses. For the FTIR spectra in figure 2, a shift of 4 cm-1 should not be relied upon for whether a reaction successfully took place. Moreover, the 1629 cm-1 peak was never eliminated as claimed in the paragraph following figure 2. For figure 3, we cannot rely on a slight diminish in peak intensity as evidence for a desired reaction successfully taking place. These FTIR spectra should be supplemented with NMR and perhaps even MS to prove that.

            In section 2.3 3D printing behavior, it is doubtful that a lower Dp results in higher commercial viability because printing times will be much slower. Since the parameters of a light source can be manipulated, a relatively high Dp can be compensated for with a lower intensity of incident light.

            In section 3.1 Materials, the iodine value for the waste cooking oil, 128.2g I2/100g is very unclear.

            Generally, the paper is close to being ready for publication but needs more chemical analysis to prove that the desired monomer is being copolymerized.

General Structural and Writing Concerns

            The draft is pretty disorganized with a lot of explanations that come before any figures and pictures to see what the paragraphs are talking about

Comments on the Quality of English Language

English is mostly ok, just some minor sentance structure and word choice issues.

Round 2

Reviewer 2 Report

Comments and Suggestions for Authors

The authors have adequately addressed the reviewer's concerns.